# Quantifying Microalgae Growth by the Optical Detection of Glucose in the NIR Waveband

**DOI:** 10.3390/molecules28031318

**Published:** 2023-01-30

**Authors:** Vimal Angela Thiviyanathan, Pin Jern Ker, Eric P. P. Amin, Shirley Gee Hoon Tang, Willy Yee, M. Z. Jamaludin

**Affiliations:** 1Institute of Sustainable Energy, Universiti Tenaga Nasional, Kajang 43000, Selangor, Malaysia; 2Center for Toxicology and Health Risk Studies (CORE), Faculty of Health Sciences, Universiti Kebangsaan Malaysia, Bangi 43600, Selangor, Malaysia; 3Faculty of Science and Marine Environment, Universiti Malaysia Terengganu, Kuala Terengganu 21030, Terengganu, Malaysia

**Keywords:** microalgae, energy, optical detection, growth monitoring, optical spectroscopy, optical monitoring, spectroscopy, glucose detection, direct detection

## Abstract

Microalgae have become a popular area of research over the past few decades due to their enormous benefits to various sectors, such as pharmaceuticals, biofuels, and food and feed. Nevertheless, the benefits of microalgae cannot be fully exploited without the optimization of their upstream production. The growth of microalgae is commonly measured based on the optical density of the sample. However, the presence of debris in the culture and the optical absorption of the intercellular components affect the accuracy of this measurement. As a solution, this paper introduces the direct optical detection of glucose molecules at 940–960 nm to accurately measure the growth of microalgae. In addition, this paper also discusses the effects of the presence of glucose on the absorption of free water molecules in the culture. The potential of the optical detection of glucose as a complement to the commonly used optical density measurement at 680 nm is discussed in this paper. Lastly, a few recommendations for future works are presented to further verify the credibility of glucose detection for the accurate determination of microalgae’s growth.

## 1. Introduction

Over the last decade, microalgae have been a popular research area owing to the wide spectrum of biomolecules that they can produce [1,2]. The application of these biomolecules in several fields—such as pharmaceuticals [3], feed [4], and biofuel [5]—has evidenced the promising future of these photosynthetic organisms. For example, the carotenoids produced by microalgae exhibit good anti-inflammatory [6,7] and anti-oxidant [8] properties that can benefit the community. Apart from that, the incorporation of microalgal biomass has been reported to enhance the immunity of animals as well as increasing their reproductive performance [9]. The advantages of microalgae—such as low sulfur emission, short life cycle, and high biomass production—have also shifted the spotlight from fossil fuels to microalgae as an alternative source of biodiesel [10,11]. At present, more research works on microalgae are directed towards efficient CO_2_ sequestration [12], treatment of sewage [13], and pigments in cosmetics [14,15].

However, to fully exploit the advantages of microalgae, further research is encouraged to focus on the optimization of microalgae biomass. The common methods used to determine microalgal biomass include measuring the dry mass of the culture [16], counting the cells using a hemocytometer [17,18,19], and measuring the optical density of the culture at 680 nm [20,21,22]. Dry mass measurement provides accurate determination of microalgal biomass, but this method is time-consuming and requires a large quantity of sample [16,23]. The measurement using a hemocytometer, on the other hand, requires an expert to manually count the cells in the hemocytometer using a microscope [24]. Though commonly practiced, the measurement using a hemocytometer is only suitable for small-scale cultivation, as it is time-consuming [25]. Furthermore, the differences in perspective of the counting personnel may introduce some inconsistency in the measurement [26,27]. This becomes more prominent when there are clumps of cells in the culture [28].

To overcome the limitation of manual cell counting using a hemocytometer, researchers have developed devices that can automatically count the cells in a sample [29,30,31]. These devices provide quick measurement results. Nevertheless, the accuracy of their measurements is still questionable as they are vulnerable to interference from extracellular debris and the internal structures of the cell itself [32]. Apart from that, the increase in cell density may result in overlapping cells, which may affect the credibility of the measurement [33].

Optical density (OD), on the other hand, can be considered only as a proxy to the biomass concentration, as the physiological state of the strains can vary over the course of their growth [34]. Therefore, the relationship between OD and biomass concentration can deviate easily. The application of spectroscopic measurement has been widely explored for the detection of various parameters of microalgae, such as chlorophyll [35], carotenoids [36], and lipids [37,38]. These research works have presented the potential of spectroscopic methods in the field of microalgae. Nevertheless, there are some drawbacks that need to be addressed to enhance the efficiency and the accuracy of the measurements. For example, the spectroscopic detection of microalgae biomolecules requires an extraction or staining procedure prior to measurement [38,39]. Apart from that, to date, there have been no reports on the spectroscopic detection of glucose in microalgal cultures. Thus, in this work, we propose the direct detection of glucose in the NIR region to complement the OD measurement at 680 nm.

## 2. Result and Discussion

Figure 1 shows the optical absorbance of NS6 (*Scenedesmus* sp.) from day 0 to day 23 represented as A0 to A23, respectively. The samples were extracted from the culture medium and were measured using an Agilent Cary5000 Spectrophotometer (Agilent, Santa Clara, CA, USA). A few peaks were observed in the range of 200–2500 nm; however, the most significant peak observed was in the range of 940–980 nm (Figure 1).

It is well established that green microalgae undergo photosynthesis to produce glucose [40,41], which is then converted into other carbohydrate components such as starch (used for energy storage in plants) [42,43], cell wall polysaccharides (such as cellulose, hemicellulose, and pectin) [44,45,46], and glycolipids [47]. Since the absorbance at 940–960 nm is observed as early as day 0 and consistently increases until day 23, this wavelength could be highly indicative of the glucose content in the culture, as glucose is the precursor of the other complex compounds (see Figure 2) [48,49,50]. Apart from that, the increasing trend could be indicative of microalgal growth, as more glucose is synthesized with the increase in the number of days from the start of growth. In addition, other works have also reported the detection of glucose in the wavelength range of 940–960 nm [51,52,53,54]. The peak in this region is said to be associated with the O-H functional group in the compound [55]. Although there are reports on the detection of starch at the region around 980 nm [56,57], it is highly possible that the authors were actually referring to the detection of glucose, as glucose monomers combine to make starch [58,59] (see Figure 2).

One may argue that the observed peak in the range of 940–960 nm could be indicative of other compounds, such as cellulose, glycolipids, or pectin, as glucose makes up these compounds. However, these are bulky compounds with different functional groups, such as carbonyl, carboxyl, and glycerol. Therefore, the presence of bulky compounds or any functional groups is expected to introduce a shift in the absorbance wavelength, as these functional groups alter the electron-withdrawing and electron-donating effects on the whole system [63,64]. For instance, the presence of glycolipids in the NIR region was observed in the range of 1100–1330 nm [65], which is further away from the reported wavelength of 940–960 nm. Apart from that, to date, there have been no reports on the optical absorbance of these compounds at 940–960 nm.

There is also a report suggesting that the presence of glucose shifts the absorption of water molecules from 975–960 nm. This is due to the hydrogen bonding between the water molecules and the glucose compound [52,66]. The two strong absorptions of water and glucose justify the irregularities in the Gaussian curve. Nevertheless, with the increase in the concentration of glucose, the Gaussian curve becomes more ideal. This is because the increasing concentration of glucose reduces the effect of water absorption in this region [52]. To further understand the absorption pattern, we compared three main points in the graph (940 nm, 960 nm, and 980 nm). The comparison is presented in Figure 3.

As shown in Figure 3, the rate of increase in absorption after day 14 becomes slower at 980 nm as compared to 940 and 960 nm. This is because as the microalgae’s growth increases with the number of days, the increasing concentration of glucose becomes more dominant in determining the absorption as compared to water. Apart from that, the increasing concentration of glucose also reduces the concentration of free water molecules (absorption at 970–980 nm), as the water molecules constantly form hydrogen bonds with the glucose molecules, resulting in a lower absorption rate. It is also important to note that water is present in both sides of the photosynthesis equation (Equation (1)) [67]. This causes a fluctuation in the concentration of water in the culture, making water an incompatible parameter to study the growth of the microalgae culture.
(1)$${\mathrm{CO}}_{2}+2H_{2}O+8-10\;\mathrm{photons}=\left[{\mathrm{CH}}_{2}O\right]+{\;H}_{2}O+{\;O}_{2}+\mathrm{waste}\;\mathrm{heat}$$

In terms of growth phases, an exponential growth phase can be observed between days 6 and 13, while the absorption between days 13 and 15 exhibits a stationary growth phase. Although ideally there should be a death phase after the stationary growth phase [68], our experimental results showed another exponential growth phase because of the utilization of the laboratory for other experimental works. This may have caused variation or uncertainty in the growth conditions, such as light intensity [69], temperature [70], and CO_2_ availability [71], resulting in a deviation from the ideal growth phases.

Based on the above discussion, there are a few areas that can be further investigated to obtain a comprehensive understanding of the peak at 940–960 nm. First, the versatility of the detection of glucose in this region (940–960 nm) across a spectrum of microalgal strains can be further explored. The results of this work clearly demonstrate the non-destructive optical detection of glucose in microalgal culture in the near-infrared waveband, and this helps in understanding the glucose production patterns in various microalgal strains.

This method has good potential as an alternative measurement technique to the conventionally used HPLC method [72]. This is because, in contrast to the HPLC method, this suggested optical method is non-destructive and its results can be obtained in real time. Nevertheless, the accuracy and consistency of this measurement method need to be further explored to understand its similarity with the HPLC method. In addition, in comparison to other glucose determination methods, such as, chromatography [73], and polarimetry [74], this optical method is more cost and time saving as it does not require any tedious preparation step.

The application of optical spectroscopy methods for determining other biomolecules of microalgae—such as chlorophyll, proteins, and lipids—has been frequently reported [75,76,77,78]. However, the reported works required a sample preparation step prior to measurement, hindering the real-time measurement of the sample. As a solution, more work is encouraged to increase the sensitivity of the measurement so that sample preparation steps such as extraction and staining can be omitted during measurement. The results reported in this work demonstrate that the determination of biomolecules of microalgae can be carried out using optical spectroscopy without sample preparation or extraction.

Second, there are several discussions on the fate of glucose after production [79,80,81]. Nevertheless, a complete understanding of the reaction pathways of glucose under different conditions (such as temperature, nutrient contents, and light exposure) has not been achieved. Hence, the study of this area will help to assess the potential of this wavelength as a proxy to the biomass concentration in the culture.

Third, the potential of this wavelength as a complement to the commonly used 680 nm [82,83] to determine the biomass of microalgae can be further investigated. It has been elaborated that the measurement at 680 nm can be significantly affected by the pigments in the sample [84,85]. This may affect the accuracy of biomass measurement. Since glucose is the byproduct of photosynthesis, the concentration of glucose should increase simultaneously with biomass. Thus, the relationship between these two wavelengths (940–960 nm and 680 nm) can be investigated to provide a more accurate determination of microalgal biomass.

## 3. Materials and Methods

The microalgal strain *Scenedesmus* sp. (NS6) used in this experiment was acquired from a lake near Nilai Spring, Malaysia [86]. The culture medium used in this experiment was the commonly used BBM solution. The materials used in the preparation of the BBM solution are listed in Table 1.

To make 1 L of BBM solution, 10 mL each of Major A and B, along with 1 mL each of EDTA-KOH, boric acid, ferric solution, and micronutrient solution, were added to 700 mL of distilled water and mixed. More distilled water was added to adjust the solution volume to 1 L, and the pH of the medium was approximately 6.0–7.0.

The prepared BBM solution was then poured into 50 mL and 100 mL conical flasks, which were sealed using a parafilm. The BBM solution was then autoclaved and left to cool. The samples were then inoculated into the 50 mL conical flask and sealed using parafilm. The conical flask was placed under LED light for two weeks. The samples were then transferred to a bigger conical flask after two weeks, and the growth of the microalgae was recorded as day 0.

The extraction of microalgal cultures was performed every 1–3 days until day 23, and the samples were measured using a 10 mm quartz cuvette. The range of the measurement, performed using a Flame UV–Vis Spectrometer produced by OceanOptics, was 188–1029 nm.

## 4. Conclusions

In this study, we investigated the applicability of the well-established glucose detection wavelength (i.e., 940–960 nm) in determining the glucose contents in microalgae. Though the abovementioned wavelength is commonly used in various fields for glucose detection, we are the first to report the use of this wavelength in optical detection for microalgae. Glucose is the main byproduct of photosynthesis, and it increases with the growth of microalgae. Therefore, it can be a good indicator of microalgal growth.

## Figures and Tables

**Figure 1 molecules-28-01318-f001:**
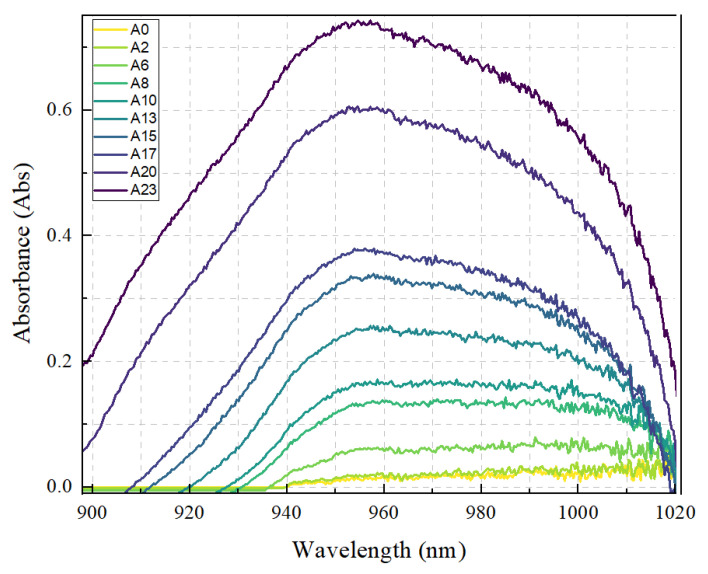
Optical absorbance of NS6 in the NIR region. The labels (A0, A1, A2, etc.) represent the days of the measurement since the start of growth.

**Figure 2 molecules-28-01318-f002:**
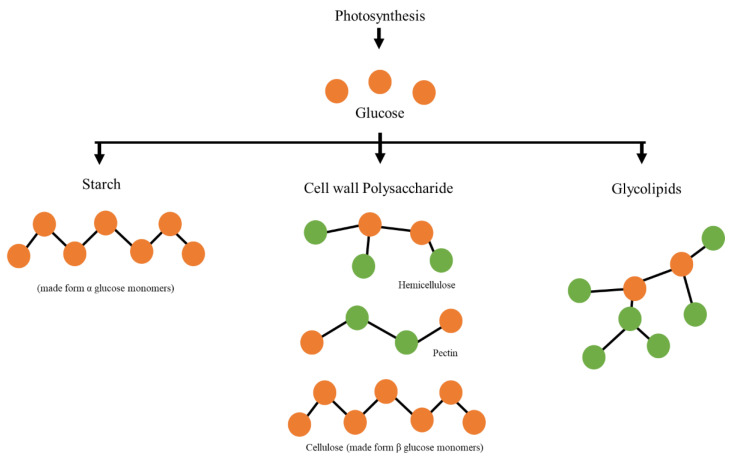
The formation of various compounds from glucose monomers. The green circles are indicative of all other compounds, such as xylose, arabinose, and glucuronic acid [60,61,62].

**Figure 3 molecules-28-01318-f003:**
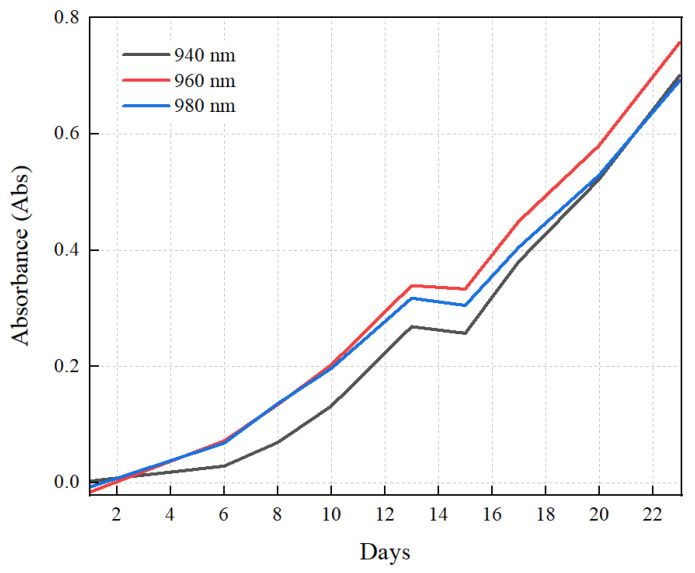
Comparison of absorbance of NS6 at 940, 960, and 980 nm as a function of the days since the start of growth.

**Table 1 molecules-28-01318-t001:** Preparation of BBM solution.

Stock Solutions and Their Solutes	Per Liter of Distilled Water
Major A	
Sodium Nitrate (NaNO_3_)	25.00 g
Calcium Chloride (CaCl_2_)	2.50 g
Magnesium Sulfate Heptahydrate (MgSO_4_.7H_2_O)	7.50 g
Sodium Chloride (NaCl)	2.50 g
Major B	
Dipotassium Phosphate (K_2_HPO_4_)	7.50 g
Monopotassium Phosphate (KH_2_PO_4_)	17.50 g
EDTA-KOH	
Ethylenediaminetetraacetic Acid (EDTA)	50.00 g
Potassium Hydroxide (KOH)	31.00 g
Ferric Solution	
Iron(II) Sulfate Heptahydrate (FeSO_4_.7H_2_O)	4.98 g
Sulfuric Acid (H_2_SO_4_)	1.00 mL
Boric Acid	
Boric Acid (H_3_BO_3_)	11.42 g
Micronutrient Solution	
Zinc Sulfate Heptahydrate (ZnSO_4_.7H_2_O)	8.82 g
Manganese(II) Chloride Tetrahydrate (MnCl_2_.4H_2_O)	1.44 g
Sodium Molybdate Dihydrate (Na_2_MoO_4_.2H_2_O	0.39 g
Copper Sulfate Pentahydrate (CuSO_4_.5H_2_O)	1.57 g
Cobalt Nitrate Hexahydrate (Co(NO_3_)_2_.6H_2_O)	0.49 g

## Data Availability

The data presented in this study are available upon reasonable request from the corresponding author.

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
