# Peer review of "Quantifying Microalgae Growth by the Optical Detection of Glucose in the NIR Waveband"

_molecules, 2023, doi:10.3390/molecules28031318_

Round 1

Reviewer 1 Report

In this work a new approach for quatification of microalgae growth is proposed. The method is based on detection of the glucose OH band at 940-960 nm in the near infrared spectrum, since the increase in glucose content is associated to microalgae growth. Detection of glucose using this approach has been previously employed for estimation of sugar content in blood, fruits, plants etc., but is the first time reported for the aforementioned purpose.

In order to test this concept, the authors recorded NIR spectra of the culture media of a microalgae strain of Scenedesmus sp. (NS6) ) at different times during 23 days. The spectra were recorded in the range between between 188 - 1029 nm and the authors focused their attention in the region between 940-980 nm (where the signal of the OH band of glucose is observed).

The manuscript is well written but the results and discussions section must be improved (see comments below). The introduction to the topic is clear and the references are pertinent; however, I recommend one additional reference to be cited (see below). The quality of the figures are good, but some corrections have to be implemented in order to improve clarity (see comments below). The methodology is clearly described. I consider that the conclusions need to be supported by additional experiments (see below). I recommend the manuscript to be reconsidered after the comments below have been fully addressed. 

Major comments:

From the plots in Figure 1 it is clear that there is an increase in the intensity of the band at 940-980 nm over time, and the authors attribute it to the increase of the glucose content over time. However, in this work, the glucose content has not been estimated used any other methods in order to validate this approach. I recommend to include data from additional experiments in order to estimated the amount of sugar content over time and compare the results with those reported herein.

In the introduction the authors mention other reference methods for quantification of microalgae growth (OD at 680 nm and hemocytometer), but they don’t report any comparison with any of those methods in order to validate the new approach. Alternative methods such as packed cell volume or dry mass can be used for comparison purpose. I recommend the authors to include at least one comparison with one of the previously reported methods.

Can the authors give an explanation of the behavior of the plots in Figure 2 between days 9-11 (the increase in the absorption becomes slightly faster) and days 11-14 (the absorption values become steady). Also note that there is an inconsistency in the x-axis ticks (there is a missing value between 10-13 days and between 17-20).

Minor comments:

Page 1, Abstract: where it reads “940 nm – 960 nm “ please remove “nm” after “940”

Page 1-2, Introduction: I recommend the authors to add the following reference in the introduction since it summarize a number of methods used for monitoring microalgae cultures:

Schagerl et al Estimating Biomass and Vitality of Microalgae for Monitoring Cultures: A Roadmap for Reliable Measurements, Cells 2022, 11, 2455. 

I recommend the authors to briefly mention alternative methods in the introduction such as packed cell volume, dry mass, etc. 

Page 2, Section Result and Discussion: This section start with the following sentence “One of the most significant peaks observed was in the range of 940 nm – 980 nm 71 (Figure 1).” There is no context of the experiment the authors are referring to at this point, and Figure 1 is not clear until reaching the experimental section a few pages below. Please add a few sentences to describe the background of the experiment you are mentioning.

Page 2, Figure 1:  The abbreviation NS6 has not been established before it is mentioned in the figure legend for first time. Also, it is no clear what the different curves represent. Please describe what the different labels (A0, A2, etc) represent (i.e. days since the growth started)

Page 3, Figure 2: Please add further information on the system the measures were performed in the figure legend and main text. The units of the absorbance in the y-axis need to be corrected. Add the units (nm) after 940, 960 and 980 in the box where the curves are labeled according to their color. Note the inconsistency in the x-axis ticks (there is a missing value between 10-13 days and between 17-20).

Author Response

Please see the attachment for specific response to reviewer 1

We are not sure which reviewer has suggested that the manuscript should undergo extensive English revisions. Therefor, foe this specific comment, we will address it here.

Author Response: We thank the reviewer for the suggestion. We have identified the grammatical errors in the manuscript and have made corrections. We have also engaged a colleague who is an English native speaker to proofread the manuscript. 

Reviewer 2 Report

This manuscript tracks the glucose signals of 940-960 nm to monitor the growth of microalgae. I believe in this way the manuscript can be improved and the scope can be broadened.

1) The basis of this work is that the 940-960 nm wave-length range is mainly associated with glucose. Currently, it is briefly described in Lines 76-81. This statement needs to be elaborated and supported by more experimental results and supporting figures. For example, could the authors show the spectrum with the full wavelength range and provide assignment of all representative signals. Also, could the authors distinguish the contribution from glucose and from other carbohydrate components such as starch, cell wall polysaccharides, and glycolipid molecules?

2) The manuscript is drafted like an experimental/lab report without sufficient discussion (not like a full research article). I would suggest the authors endeavor to better correlating the findings with the existing studies of the algal research field.

3) Another weak aspect is the lack of cross comparing the method with other characterization and quantification techniques. For example, cellular NMR methods have been employed to closely monitor the content and composition of algal biomolecules. A few representative studies are provided below and there are many other relevant studies in the field.

This is a study quantifying the glycan molecules using intact microalgal cells: Poulhazan et al. Identification and Quantification of Glycans in Whole Cells: Architecture of Microalgal Polysaccharides Described by Solid-State NMR. J. Am. Chem. Soc. 143, 46, 19374-388 (2021).

This is a recent review including a summary of the extensive cellular NMR investigations of algal cells as well as different models of their extracellular matrices: Ghassemi et al. Solid-State NMR Investigations of Extracellular Matrices and Cell Walls of Algae, Bacteria, Fungi, and Plants. Chem. Rev. 122, 10036−10086 (2022).

4) It might be helpful to integrate the concepts of carbohydrate polymers in the above references to the discussion of this manuscript. Currently, this aspect is missing.

5) Another comment related to the detection of optical signals: would the authors add a discussion regarding if the method could be applied to the detection of other biomolecules in the algal cells, besides glucose?

Author Response

Please see the attachment for specific response to reviewer 2

We are not sure which reviewer has suggested that the manuscript should undergo extensive English revisions. Therefor, for this specific comment, we will address it here.

Author Response: We thank the reviewer for the suggestion. We have identified the grammatical errors and have made corrections. We have also engaged a colleague who is an English native speaker to proofread the manuscript. 

Round 2

Reviewer 1 Report

Thanks to the authors for addressing all the questions. 

I am still having concerns about a few statements:

Page 4, lines 92-110: What are the different functional grupos the authors are referring to in lines 106-107? Since each glucose residue in starch, cellulose, etc, still contains three free hydroxyl  grups at positions 2, 3 and 6, how are these signals supposed to be are affected? Furthermore the other monosaccharides (colored in green in the representations of Figure 2) also contain the same funcional group. Could the authors please explain further this statement in order to address this question and mention a reference where spectra of some of the aforementioned polymers have been reported? Alternatively, have the authors considered  acquiring  a reference spectrum of starch in order to confirm that there is not contribution of these compounds in the signal observed in the region between 940 and 960 nm?

Page 5, line 135-137:  How the factors listened in line 136 (light intensity, temperature and CO2 availability) influenced the experiment carried out by the authors? Are there any changes in those conditions after day 15 that need to be reported?  Could you please be more specific concerning the last sentence. What kind of noise you the authors referring to in line 137? 

Some minor comments:

  1. Page 1, line 13-14: note the the word “sector” is mentioned 4 times in the same sentence.
  2. Page 5, Line 120: remove the additional “nm” after “940” and “960”. Consider adding the sentence “of NS6 as a function of the days since the growth started” at the end of the sentence.
  3. Page 5, Line 122: remove the additional “nm” after “940”
